# Learning Bounds for Greedy Approximation with Explicit Feature Maps from Multiple Kernels

**Shahin Shahrampour**
Department of Industrial and Systems Engineering
Texas A&M University
College Station, TX 77843
shahin@tamu.edu

**Vahid Tarokh**
Department of Electrical and Computer Engineering
Duke University
Durham, NC 27708
vahid.tarokh@duke.edu

## Abstract

Nonlinear kernels can be approximated using finite-dimensional feature maps for efficient risk minimization. Due to the inherent trade-off between the dimension of the (mapped) feature space and the approximation accuracy, the key problem is to identify promising (explicit) features leading to a satisfactory out-of-sample performance. In this work, we tackle this problem by efficiently choosing such features from multiple kernels in a greedy fashion. Our method sequentially selects these explicit features from a set of candidate features using a correlation metric. We establish an out-of-sample error bound capturing the trade-off between the error in terms of explicit features (approximation error) and the error due to spectral properties of the best model in the Hilbert space associated to the combined kernel (spectral error). The result verifies that when the (best) underlying data model is sparse enough, i.e., the spectral error is negligible, one can control the test error with a small number of explicit features, that can scale poly-logarithmically with data. Our empirical results show that given a fixed number of explicit features, the method can achieve a lower test error with a smaller time cost, compared to the state-of-the-art in data-dependent random features.

## 1   Introduction

Kernel methods are powerful tools in describing the nonlinear representation of data. Mapping the inputs to a high-dimensional feature space, kernel methods compute their inner products without recourse to the explicit form of the *feature map* (kernel trick). However, unfortunately, calculating the kernel matrix for the training stage requires a prohibitive computational cost scaling quadratically with data. To address this shortcoming, recent years have witnessed an intense interest on the approximation of kernels using low-rank surrogates [1, 2, 3]. Such techniques can turn the kernel formulation to a linear problem, which is potentially solvable in a linear time with respect to data (see e.g. [4] for linear Support Vector Machines (SVM)) and thus applicable to large data sets. In the approximation of kernels via their corresponding finite-dimensional feature maps, regardless of whether the approximation is deterministic [5] or random [3], it is extremely critical that – we can compute the feature maps efficiently – and – we can (hopefully) represent the data in a sparse fashion. The challenge is that finding feature maps with these characteristics is generally hard.

It is well-known that any Mercer kernel can be represented as an (potentially infinite-dimensional) inner-product of its feature maps, and thus, it can be approximated with an inner product in a lower dimension. As an example, the explicit feature map (also called Taylor feature map) of the Gaussian kernel is derived in [6] via Taylor expansion. In supervised learning, the key problem is to identify the explicit features [1] that lead to low out-of-sample error as there is an inherent trade-off between the computational complexity and the approximation accuracy. This will turn the learning problem at hand into an optimization with sparsity constraints, which is is generally NP-hard.

In this paper, our objective is to present a method for efficiently "choosing" explicit features associated to a number of base positive semi-definite kernels. Motivated by the success of greedy methods in sparse approximation [7, 8], we propose a method to select promising features from multiple kernels in a greedy fashion. Our method, dubbed Multi Feature Greedy Approximation (MFGA), has access to a set of candidate features. Exploring these features sequentially, the algorithm maintains an active set and adds one explicit feature to it per step. The selection criterion is according to the correlation of the gradient of the empirical risk with the standard bases.

We provide non-asymptotic guarantees for MFGA, characterizing its out-of-sample performance via three types of errors, one of which (spectral error) relates to spectral properties of the best model in the Hilbert space associated to the combined kernel. Our theoretical result suggests that if the underlying data model is sparse enough, i.e., the spectral error is negligible, one can achieve a low out-of-sample error with a small number of features, that can scale poly-logarithmically with data. Recent findings in [9] shows that in approximating square integrable functions with smooth radial kernels, the coefficient decay is nearly exponential (small spectral error). In light of these results, our method has potential in constructing sparse representations for a rich class of functions.

We further provide empirical evidence (Section 5) that explicit feature maps can be efficient tools for sparse representation. In particular, compared to the state-of-the-art in data-dependent random features, MFGA requires a smaller number of features to achieve a certain test error on a number of datasets, while spending less computational resource. Our work is related to several lines of research in the literature, namely random and deterministic kernel approximation, sparse approximation, and multiple kernel learning. Due to variety of these works, we postpone the detailed discussion of the related literature to Section 4, after presenting the preliminaries, formulation, and results.

## 2 Problem Formulation

**Preliminaries:** Throughout the paper, the vectors are all in column format. We denote by $[N]$ the set of positive integers $\{1, \ldots, N\}$, by $\langle \mathbf{x}, \mathbf{x}' \rangle$ the inner product of vectors $\mathbf{x}$ and $\mathbf{x}'$ (in potentially infinite dimension), by $\|\cdot\|_p$ the $p$-norm operator, by $\mathcal{L}^2(\mathcal{X})$ the set of square integrable functions on the domain $\mathcal{X}$, and by $\Delta_P$ the $P$-dimensional probability simplex, respectively. The support of vector $\boldsymbol{\theta} \in \mathbb{R}^d$ is $\operatorname{supp}(\boldsymbol{\theta}) \triangleq \{i \in [d] : \theta_i \neq 0\}$. $\lceil \cdot \rceil$ and $\lfloor \cdot \rfloor$ denote the ceiling and floor functions, respectively. We make use of the following definitions:

**Definition 1.** *(strong convexity) A differentiable function $g(\cdot)$ is called $\mu$-strongly convex on the domain $\mathcal{X}$ with respect to $\|\cdot\|_2$, if for all $\mathbf{x}, \mathbf{x}' \in \mathcal{X}$ and some $\mu > 0$,*

$$g(\mathbf{x}) \geq g(\mathbf{x}') + \langle \nabla g(\mathbf{x}'), \mathbf{x} - \mathbf{x}' \rangle + \frac{\mu}{2} \|\mathbf{x} - \mathbf{x}'\|_2^2 .$$

**Definition 2.** *(smoothness) A differentiable function $g(\cdot)$ is called $\beta$-smooth on the domain $\mathcal{X}$ with respect to $\|\cdot\|_2$, if for all $\mathbf{x}, \mathbf{x}' \in \mathcal{X}$ and some $\beta > 0$,*

$$g(\mathbf{x}) \leq g(\mathbf{x}') + \langle \nabla g(\mathbf{x}'), \mathbf{x} - \mathbf{x}' \rangle + \frac{\beta}{2} \|\mathbf{x} - \mathbf{x}'\|_2^2 .$$

### 2.1 Supervised Learning with Explicit Feature Maps

In supervised learning, a training set $\{(\mathbf{x}_n, y_n)\}_{n=1}^N$ in the form of *input-output* pairs is given to the learner. The (input-output) samples are generated independently from an *unknown* distribution $P_{\mathcal{X}\mathcal{Y}}$. For $n \in [N]$, we have $\mathbf{x}_n \in \mathcal{X} \subset \mathbb{R}^d$. In the case of regression, the output variable $y_n \in \mathcal{Y} \subseteq [-1, 1]$,

whereas in the case of classification $y_n \in \{-1, 1\}$. The ultimate objective is to find a target function $f : \mathcal{X} \to \mathbb{R}$, to be employed in mapping (unseen) inputs to correct outputs. This goal may be achieved through minimizing a risk function $R(f)$, defined as

$$R(f) \triangleq \mathbb{E}_{P_{\mathcal{X}\mathcal{Y}}}[L(f(\mathbf{x}), y)] \qquad \widehat{R}(f) \triangleq \frac{1}{N} \sum_{n=1}^{N} L(f(\mathbf{x}_n), y_n), \qquad (1)$$

where $L(\cdot, \cdot)$ is a loss function depending on the task (e.g., quadratic for regression, hinge loss for SVM). Since the distribution $P_{\mathcal{X}\mathcal{Y}}$ is unknown, in lieu of the true risk $R(f)$, we minimize the empirical risk $\widehat{R}(f)$. To solve the problem, one needs to consider a function class for $f(\cdot)$ to minimize the empirical risk over that class. For example, consider a positive semi-definite kernel $K(\cdot, \cdot)^2$ and consider functions of the form $f(\cdot) = \sum_{n=1}^{N} \alpha_n K(\mathbf{x}_n, \cdot)$. Kernel methods minimize the empirical risk $\widehat{R}(f)$ over this class of functions by solving for optimal values of parameters $\{\alpha_n\}_{n=1}^{N}$. While being theoretically well-justified, this approach is not practically applicable to large datasets, as $O(N^2)$ computations are required just to set up the training problem.

We now face two important questions: (i) can we reduce the computation time using a suitable approximation of the kernel? (ii) how does the choice of kernel affect the prediction of unseen data (generalization performance)? There is a large body of literature addressing these two questions. We provide an extensive discussion of the related works in Section 4, and here, we focus on presenting our method aiming to tackle the challenges above.

Consider a set of base positive semi-definite kernels $\{K_1, \ldots, K_P\}$, such that $K_p(\mathbf{x}, \mathbf{x}') = \langle \boldsymbol{\phi}_p(\mathbf{x}), \boldsymbol{\phi}_p(\mathbf{x}') \rangle$ for $p \in [P]$. The feature map $\boldsymbol{\phi}_p : \mathbf{x} \mapsto \mathcal{F}_{K_p}$ maps the points in $\mathcal{X}$ to $\mathcal{F}_{K_p}$, the associated Reproducing Kernel Hilbert Space (RKHS) to kernel $K_p$. Let $\boldsymbol{\theta} = [\theta_{1,1}, \ldots, \theta_{1,M_1} \ldots, \theta_{P,1}, \ldots, \theta_{P,M_P}]^\top$ and $\boldsymbol{\nu} = [\nu_1, \ldots, \nu_P]^\top$, such that $\sum_{p=1}^{P} M_p = M$. Define

$$\widehat{\mathcal{F}}_M \triangleq \left\{ f(\mathbf{x}) = \sum_{p=1}^{P} \sum_{m=1}^{M_p} \theta_{p,m} \sqrt{\nu_p} \boldsymbol{\phi}_{p,m}(\mathbf{x}) : \|\boldsymbol{\theta}\|_2 \leq C , \ \boldsymbol{\nu} \in \Delta_P , \ \sum_{p=1}^{P} M_p = M \right\}, \quad (2)$$

where $\boldsymbol{\phi}_{p,m}(\cdot)$ is the $m$-th component of the explicit feature map associated to $K_p$. The use of explicit feature maps has proved to be beneficial in learning with significantly smaller computational burden (see e.g. [6] for approximation of Gaussian kernel in training SVM and [5] for explicit form of feature maps for several practical kernels). We use $\boldsymbol{\nu}$ for normalization purposes, and we are *not* concerned with learning a rule to optimize it. Instead, given a fixed value of $\boldsymbol{\nu}$, we are interested in including promising $\boldsymbol{\phi}_{p,m}(\cdot)$'s in $\widehat{\mathcal{F}}_M$, i.e., the ones improving generalization. We can always optimize the performance over $\boldsymbol{\nu}$. It is actually well-known that Multiple Kernel Learning (MKL) can potentially improve the generalization; however, it comes at the cost of solving expensive optimization problems [10].

Note that the set $\widehat{\mathcal{F}}_M$ is a rich class of functions. It consists of $M$-term approximations of the class

$$\mathcal{F} \triangleq \left\{ f(\mathbf{x}) = \sum_{m=1}^{\infty} \sum_{p=1}^{P} \theta_{p,m} \sqrt{\nu_p} \boldsymbol{\phi}_{p,m}(\mathbf{x}) : \sum_{m=1}^{\infty} \sum_{p=1}^{P} \theta_{p,m}^2 \leq C , \ \boldsymbol{\nu} \in \Delta_P \right\}, \qquad (3)$$

using multiple feature maps. Focusing on one kernel ($P = 1$), we know by Parseval's theorem [11] that for a function in $\mathcal{L}^2(\mathcal{X})$ the $i$-th coefficient must decay faster than $O(1/\sqrt{i})$ when the bases are orthonormal. Interestingly, it has recently been proved that for approximation with smooth radial kernels, the coefficient decay is nearly exponential [9]. Therefore, for functions in $\mathcal{L}^2(\mathcal{X})$, most of the energy content comes from the initial coefficients, and we can hope to keep $M \ll N$ for computationally efficient training. Such solutions also offer $O(M)$ computations in the test phase as opposed to $O(N)$ in traditional kernel methods.

## 2.2 Multi Feature Greedy Approximation

We now propose an algorithm that carefully chooses the (approximated) kernel to attain a low out-of-sample error. The algorithm has access to a set of $M_0$ candidate (explicit) features $\boldsymbol{\phi}_{p,m}(\cdot)$,

i.e., $\sum_{p,m} 1 = M_0$. Starting with an empty set, it maintains an active set of selected features by exploring the candidate features. At each step, the algorithm calculates the correlation of the gradient (of the empirical risk) with standard bases of $\mathbb{R}^{M_0}$. The feature $\phi_{p,m}(\cdot)$ whose index coincides with the most absolute correlation is added to the active set, and next, the empirical risk is minimized over a more general model including the chosen feature. In the case of regression, if we let $\psi_{p,m}^\top = [\phi_{p,m}(\mathbf{x}_1) \cdots \phi_{p,m}(\mathbf{x}_N)]$, the algorithm selects a $\phi_{p,m}$ such that $\psi_{p,m}$ has the largest absolute correlation with the residual (the method is known as Orthogonal Matching Pursuit (OMP) [12, 13]). The algorithm can proceed for $M$ rounds or until a termination condition is met (e.g. the risk is small enough). Denoting by $\mathbf{e}_j$ the $j$-th standard basis in $\mathbb{R}^{M_0}$, we outline the method in Algorithm 1.

---

**Algorithm 1** Multi Feature Greedy Approximation (MFGA)

---

**Initialize:** $I^{(1)} = \emptyset$, $\boldsymbol{\theta}^{(0)} = \mathbb{0} \in \mathbb{R}^{M_0}$
1: **for** $t \in [M]$ $(M < M_0)$ **do**
2:     Let $J^{(t)} = \text{argmax}_{j \in [M_0]} \left| \left\langle \nabla \widehat{R} \left( \boldsymbol{\theta}^{(t-1)} \right), \mathbf{e}_j \right\rangle \right|$.
3:     Let $I^{(t+1)} = I^{(t)} \cup \{J^{(t)}\}$.
4:     Solve $\boldsymbol{\theta}^{(t)} = \text{argmin}_{f \in \widehat{\mathcal{F}}_{M_0}} \{\widehat{R}(f)\}$ subject to $\text{supp}(\boldsymbol{\theta}) = I^{(t+1)}$.
5: **end for**
**Output:** $\widehat{f}_{\text{MFGA}}(\cdot) = \sum_{p=1}^{P} \sum_{m=1}^{M_p} \theta_{p,m}^{(M)} \sqrt{\nu_p} \phi_{p,m}(\cdot)$.

---

Assuming that repetitive features are not selected, at each iteration of the algorithm, a linear regression or classification is solved over a variable of size $t$. If the time cost of the task is $\mathcal{C}(t)$, the training cost of MFGA would be $\sum_{t=1}^{M} \mathcal{C}(t)$. However, in practice, we can select multiple features at each iteration to decrease the runtime of the algorithm. In the case of regression, this amounts to Generalized OMP [14]. While in general this rule might be sub-optimal, the authors of [14] have shown that the method is quite competitive to the original OMP where one element is selected per iteration.

## 3 Theoretical Guarantees

Recall that our objective is to evaluate the out-of-sample performance (generalization) of our proposed method. To begin, we quantify the richness of the class (2) in Lemma 1 using the notion of Rademacher complexity, defined below:

**Definition 3.** *(Rademacher complexity) For a finite-sample set $\{\mathbf{x}_i\}_{i=1}^{N}$, the empirical Rademacher complexity of a class $\mathcal{F}$ is defined as*

$$\widehat{\mathcal{R}}(\mathcal{F}) \triangleq \frac{1}{N} \mathbb{E}_{\mathcal{P}_\sigma} \left[ \sup_{f \in \mathcal{F}} \sum_{i=1}^{N} \sigma_i f(\mathbf{x}_i) \right],$$

*where the expectation is taken over $\{\sigma_i\}_{i=1}^{N}$ that are independent samples uniformly distributed on the set $\{-1, 1\}$. The Rademacher complexity is then $\mathcal{R}(\mathcal{F}) \triangleq \mathbb{E}_{\mathcal{P}_\mathcal{X}} \widehat{\mathcal{R}}(\mathcal{F})$.*

**Assumption 1.** *For all $p \in [P]$, $K_p$ is a positive semi-definite kernel and $\sup_{\mathbf{x} \in \mathcal{X}} K_p(\mathbf{x}, \mathbf{x}) \leq B^2$.*

**Lemma 1.** *Given Assumption 1, the Rademacher complexity of the function class (2) is bounded as,*

$$\mathcal{R}(\widehat{\mathcal{F}}_M) \leq BC \sqrt{\frac{3 \lceil \log P \rceil}{N}}.$$

The bound above exhibits mild dependence to the number of base kernels $P$, akin to the results in [15]. To derive our theoretical guarantees, we rely on the following assumptions:

**Assumption 2.** *The loss function $L(y, y') = L(yy')$ is $\beta$-smooth and $G$-Lipschitz in the first argument.*

Notable example of the loss function satisfying the assumption above is the logistic loss $L(y, y') = \log(1 + \exp(-yy'))$ for binary classification.

**Assumption 3.** *The empirical risk $\widehat{R}$ is $\mu$-strongly convex with respect to $\boldsymbol{\theta}$.*

In case the empirical risk is weakly convex, strongly convexity can be achieved via adding a Tikhonov regularizer. We are now ready to present our main theoretical result which decomposes the out-of-sample error into three components:

**Theorem 2.** *Define $f^\star(\cdot) \triangleq argmin_{f \in \mathcal{F}} R(f) = \sum_{p=1}^{P} \sum_{m=1}^{\infty} \theta_{p,m}^\star \sqrt{\nu_p} \phi_{p,m}(\cdot)$. Let Assumptions 1-3 hold and $\boldsymbol{\theta}^{(t)} \in \{\boldsymbol{\theta} \in \mathbb{R}^{M_0} : \|\boldsymbol{\theta}\|_2 < C\}$ for $t \in [M]$. Then, after $M$ iterations of Algorithm 1, the output satisfies,*

$$R(\widehat{f}_{MFGA}) - \min_{f \in \mathcal{F}} R(f) \leq \mathcal{E}_{est} + \mathcal{E}_{app} + \mathcal{E}_{spec},$$

*with probability at least $1 - \delta$ over data, where*

$$\mathcal{E}_{est} = O\left( \frac{\sqrt{\lceil \log P \rceil} + \sqrt{-\log \delta}}{\sqrt{N}} \right), \quad \mathcal{E}_{app} = O\left( \exp\left( -\lfloor M^{1-\varepsilon} \rfloor \left\lceil \frac{\beta}{\mu} \right\rceil^{-1} \right) \right), \quad \mathcal{E}_{spec} = O\left( \sqrt{\sum_{p=1}^{P} \sum_{m=\lfloor \frac{\lfloor M^\varepsilon \rfloor}{P} \rfloor}^{\infty} {\theta_{p,m}^\star}^2} \right),$$

*for any $\varepsilon \in (0, 1)$.*

Our error bound consists of three terms: estimation error $\mathcal{E}_{\text{est}}$, approximation error $\mathcal{E}_{\text{app}}$, and spectral error $\mathcal{E}_{\text{spec}}$. As the bound holds for $\varepsilon \in (0, 1)$, it can optimized over the choice of $\varepsilon$ in theory. The $O(1/\sqrt{N})$ estimation error with respect to the sample size is quite standard in supervised learning. It was also shown in [15] that one cannot improve upon the $\sqrt{\log P}$ dependence due to the selection of multiple kernels. The approximation error shows that the decay is exponential with respect to the number of features, i.e., to get an $O(1/\sqrt{N})$ error, we only need $O((\log N)^{\frac{1}{1-\varepsilon}})$ features. The exponential decay (expected from the greedy methods [8, 16]) dramatically reduces the number of features compared to non-greedy, randomized techniques at the cost of more computation. The "spectral" error characterizes the spectral properties of the best model in the class (3). Since the 2-norm of the coefficient sequence is bounded, $\mathcal{E}_{\text{spec}} \to 0$ as $M \to \infty$, but the rate depends on the tail of the coefficient sequence. For example, if for all $p \in [P]$, $K_p$ is a smooth radial kernel, the coefficient decay is nearly exponential [9].

**Remark 1.** *The quadratic loss $L(y, y') = (y - y')^2$ does not satisfy Assumption 2 in the sense that $L(y, y') \neq L(yy')$, but with similar analysis in Theorem 2, we can prove that the same error bound holds with slightly different constants (see the supplementary material).*

**Remark 2.** *Using Theorem 2.8 in [17], our result can be extended to $\ell_2$-regularized risk (see [17], Remark 2.1). In case of $\ell_1$-penalty, due to non-differentiability, we should work with alternatives (e.g. $\log[\cosh(\cdot)]$).*

**Remark 3.** *There is an interesting connection between our result and reconstruction bounds in greedy methods (e.g. [8]), where using $M$ bases, the error decay is a function of both $M$ and "the best reconstruction" with $M$ bases. Similarly here, $\mathcal{E}_{app}$ and $\mathcal{E}_{spec}$ capture these two notions, respectively. Both errors go to zero as $M \to \infty$ and there is a trade-off between the two, given $\varepsilon > 0$. An important issue is that "the best reconstruction" depends on the initial candidate (explicit features) set. That error is small if the good explicit features are in the candidate set, and in a Fourier analogy, a signal should be "band-limited" to be approximated well with finite bases.*

## 4 Related Literature

Our work is related to several strands of literature reviewed below:

**Kernel approximation:** Since the kernel matrix is $N \times N$, the computational cost of kernel methods scales at least quadratically with respect to data. To overcome this problem, a large body of literature has focused on approximation of kernels using low-rank surrogates [1, 2]. Examples include the celebrated Nyström method [18, 19] which samples a subset of training data, approximates a surrogate kernel matrix, and then transforms the data using the approximated kernel. Shifting focus to explicit feature maps, in [20, 21], the authors have proposed low-dimensional Taylor expansions of Gaussian kernel for speeding up learning. Moreover, Vedaldi et al. [22] provide explicit feature maps for additive homogeneous kernels and quantify the approximation error using this approach. The major

difference of our work with this literature is that we are concerned with *selecting "good" feature maps* in a greedy fashion for improved generalization.

**Random features:** An elegant idea to improve the efficiency of kernel approximation is to use randomized features [3, 23]. In this approach, the kernel function can be approximated as

$$K(\mathbf{x}, \mathbf{x}') = \int_{\Omega} \phi(\mathbf{x}, \boldsymbol{\omega}) \phi(\mathbf{x}', \boldsymbol{\omega}) dP_{\Omega}(\boldsymbol{\omega}) \approx \frac{1}{M} \sum_{m=1}^{M} \phi(\mathbf{x}, \boldsymbol{\omega}_m) \phi(\mathbf{x}', \boldsymbol{\omega}_m), \tag{4}$$

using Monte Carlo sampling of *random features* $\{\boldsymbol{\omega}_m\}_{m=1}^{M}$ from the support set $\Omega$. A wide variety of kernels can be written in the form of above. Examples include shift-invariant kernels approximated by Monte Carlo [3] or Quasi Monte Carlo [24] sampling as well as dot product (e.g. polynomial) kernels [25]. Various methods have been developed to decrease the time and space complexity of kernel approximation (see e.g. Fast-food [26] and Structured Orthogonal Random Features [27]) using properties of dense Gaussian random matrices. In general, random features reduce the computational complexity of traditional kernel methods. It has been shown recently in [28] that to achieve $O(1/\sqrt{N})$ learning error, we require only $M = O(\sqrt{N} \log N)$ random features. Also, the authors of [29] have shown that by $\ell_1$-regularization (using a randomized coordinate descent approach) random features can be made more efficient. In particular, to achieve $\epsilon$-precision on risk, $O(1/\epsilon)$ random features would be sufficient (as opposed to $O(1/\epsilon^2)$).

Another line of research has focused on *data-dependent* choice of random features. In [30, 31, 32, 33], data-dependent random features has been studied for the approximation of shift-invariant/translation-invariant kernels. On the other hand, in [34, 35, 36, 37], the focal point is on the improvement of the out-of-sample error. Sinha and Duchi [34] propose a pre-processing optimization to re-weight random features, whereas Shahrampour et al. [35] introduce a data-dependent score function to select random features. Furthermore, Bullins et al. [37] focus on approximating translation-invariant/rotation-invariant kernels and maximizing kernel alignment in the Fourier domain. They provide analytic results on classification by solving the SVM dual with a no-regret learning scheme, and also an improvement is achieved in terms of using multiple kernels. The distinction of our work with this literature is that our method is greedy rather than randomized, and our focus is on explicit feature maps. Additionally, another significant difference in our framework with that of [37] is that we work with differentiable loss functions, whereas [37] focuses on SVM. We will compare our work with [23, 34, 35].

**Greedy approximation:** Over the pas few decades, greedy methods such as Matching Pursuit (MP) [38, 7] and Orthogonal Matching Pursuit (OMP) [12, 13, 8] have attracted the attention of several communities due to their success in sparse approximation. In the machine learning community, Vincent et al. [39] have proposed MP and OMP with kernels as elements. In the similar spirit is the work of [40], which concentrates on sparse regression and classification models using Mercer kernels, as well as the work of [41] that considers sparse regression with multiple kernels. Though traditional MP and OMP were developed for regression, they have been further extended to logistic regression [42] and smooth loss functions [43]. Moreover, in [44], a greedy reconstruction technique has been developed for regression by empirically fitting squared error residuals. Unlike most of the prior art, our focus is on explicit feature maps rather than kernels to save significant computational costs. Our algorithm can be thought as an extension of fully corrective greedy in [17] to nonlinear features from multiple kernels where we optimize the risk over the class (2). However, in MFGA, we work with the empirical risk (rather than the true risk in [17]), which happens in practice as we do not know $P_{\mathcal{X}\mathcal{Y}}$.

**Multiple kernel learning:** The main objective of MKL is to identify a good kernel using a data-dependent procedure. In supervised learning, these methods may consider optimizing a convex, linear, or nonlinear combination of a number of base kernels with respect to some measure (e.g. kernel alignment) to select an ideal kernel [45, 46, 47]. It is also possible to optimize the kernel as well as the empirical risk simultaneously [48, 49]. On the positive side, there are many theoretical guarantees for MKL [15, 50], but unfortunately, these methods often involve computationally expensive steps, such as eigen-decomposition of the Gram matrix (see [10] for a comprehensive survey). The major difference of this work with MLK is that we consider a combination of explicit feature maps (rather than kernels), and more importantly, we do *not* optimize the weights (as mentioned in Section 2.1, we do not optimize the class (2) over $\boldsymbol{\nu}$) to avoid computational cost. Instead, our goal is to *greedily* choose promising features for a fixed value of $\boldsymbol{\nu}$.

We finally remark that data-dependent learning has been explored in the context of boosting and deep learning [51, 52, 53]. Here, our main focus is on sparse representation for shallow networks.

# 5 Empirical Evaluations

We now evaluate our method on several datasets from the UCI Machine Learning Repository.
**Benchmark algorithms:** We compare MFGA to the state-of-the-art in randomized kernel approximation as well as traditional kernel methods:
**1) RKS [23]**, with approximated Gaussian kernel: $\phi = \cos(\mathbf{x}^\top \boldsymbol{\omega}_m + b_m)$ in (4), $\{\boldsymbol{\omega}_m\}_{m=1}^M$ are sampled from a Gaussian distribution, and $\{b_m\}_{m=1}^M$ are sampled from the uniform distribution on $[0, 2\pi)$.
**2) LKRF [34]**, with approximated Gaussian kernel: $\phi = \cos(\mathbf{x}^\top \boldsymbol{\omega}_m + b_m)$ in (4), but instead of $M$, a larger number $M_0$ random features are sampled and then re-weighted by solving a kernel alignment optimization. The top $M$ random features would be used in the training.
**3) EERF [35]**, with approximated Gaussian kernel: $\phi = \cos(\mathbf{x}^\top \boldsymbol{\omega}_m + b_m)$ in (4), and again $M_0$ random features are sampled and then re-weighted according to a score function. The top $M$ random features would appear in the training. See Table 2a-2b for values of $M$ and $M_0$.
**4) GK,** the standard Gaussian kernel.
**5) GLK,** which is a sum of a Gaussian and a linear kernel.

The selection of the baselines above allows us to investigate the time-vs-accuracy tradeoff in kernel approximation. Ideally, we would like to outperform randomized approaches, while being competitive to kernel methods with significantly lower computational cost.

**Practical considerations:** To determine the width of the Gaussian kernel $K(\mathbf{x}, \mathbf{x}') = \exp(-\|\mathbf{x} - \mathbf{x}'\|^2 / 2\sigma^2)$, we choose the value of $\sigma$ for each dataset to be the mean distance of the 50th $\ell_2$ nearest neighbor. Though being a rule-of-thumb, this choice has exhibited good generalization performance [30]. Notice that for randomized approaches, this amounts to sampling random features from $\sigma^{-1}\mathcal{N}(0, I_d)$. Of course, optimizing over $\sigma$ (e.g. using cross-validation, jackknife, or their approximate surrogates [54, 55, 56]) may provide better results. For our method as well as GLK, we do not optimize over the convex combination weights (uniform weights are assigned). This is possible using MKL, but our goal is to evaluate the trade-off between approximation and accuracy, rather than proposing a rule to learn the best possible weights for the kernel. For classification, we let the number of candidate features $M_0 = 2d + 1$, consisting of the first order Taylor features of the Gaussian kernel combined with features of linear kernel, whereas for regression, we let $M_0 = \binom{d}{2} + 2d + 1$, approximating the Gaussian kernel up to second order. In the experiments, we replace the 2-norm constraint of (2) by a quadratic regularizer [23], tune the regularization parameter over the set $\{10^{-5}, 10^{-4}, \dots, 10^5\}$, and report the best result for each method. As noted in Section 2.2, we select multiple features at each iteration of MFGA which is suboptimal but decreases the runtime of the algorithm. We use logistic regression model for classification to be able to compute the gradient needed in MFGA.

**Datasets:** In Table 1, we report the number of training samples ($N_{\text{train}}$) and test samples ($N_{\text{test}}$) used for each dataset. If the training and test samples are not provided separately for a dataset, we split it randomly. We standardize the data in the following sense: we scale the features to have zero mean and unit variance and the responses in regression to be inside $[-1, 1]$.

Table 1: Input dimension, number of training samples, and number of test samples are denoted by $d$, $N_{\text{train}}$, and $N_{\text{test}}$, respectively.

| Dataset | Task | $d$ | $N_{\text{train}}$ | $N_{\text{test}}$ |
|---|---|---|---|---|
| Year prediction | Regression | 90 | 46371 | 5163 |
| Online news popularity | Regression | 58 | 26561 | 13083 |
| Adult | Classification | 122 | 32561 | 16281 |
| Epileptic seizure recognition | Classification | 178 | 8625 | 2875 |

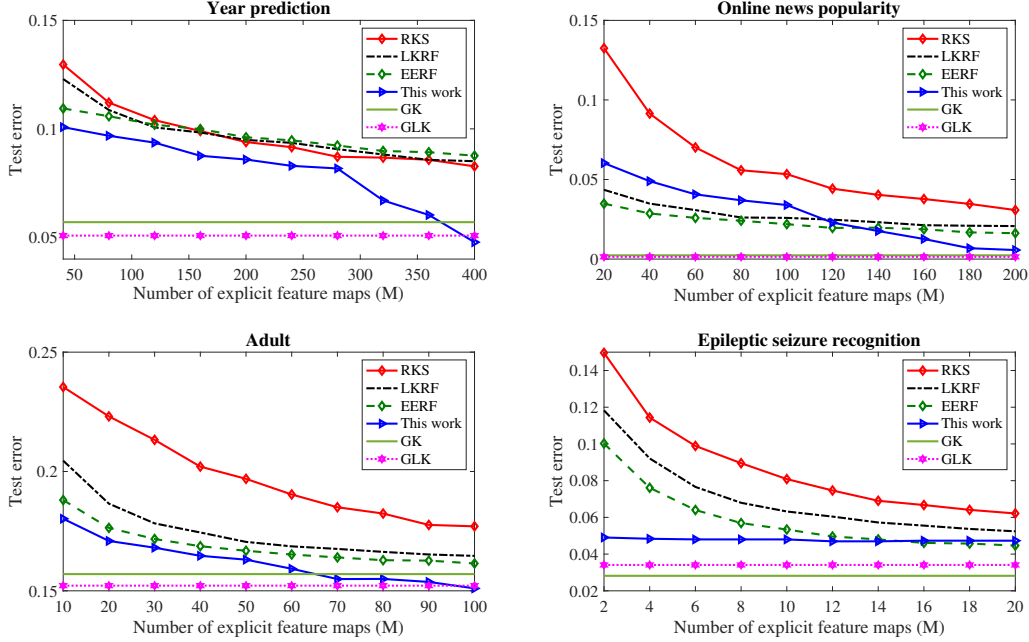

Figure 1: Comparison of the test error of MFGA (this work) versus the randomized features baselines RKS, LKRF, and EERF, as well as Gaussian Kernel (GK) and Gaussian+Linear Kernel (GLK).

**Comparison with random features:** For datasets in Table 1, we report our empirical findings in Figure 1. On "Year prediction" and "Adult", our method consistently improves the test error compared to the state-of-the-art, i.e., MFGA requires smaller number of features to achieve a certain test error threshold. The key is to select "good" features to learn the subspace, and MFGA does so by greedily searching among the candidate features that are explicit feature maps of the linear+Gaussian kernel (up to second order Taylor expansion). As the number of features $M$ increases, all methods tend to generalize better in the regime shown in Figure 1. On "Online news popularity" our method eventually achieves a smaller test error, whereas on "Epileptic seizure recognition" it is superior for $M \leq 14$ while being dominated by EERF afterwards.

Table 2a-2b tabulates the test error and time cost for largest $M$ (for each dataset) in Figure 1. Since RKS is *fully randomized* and *data-independent*, it has the smallest training time. However, in order to compare the time cost of LKRF, EERF, and our work, we need additional details as the comparison may not be immediate. In the pre-processing phase, LKRF and EERF draw $M_0$ samples from the Gaussian distribution and incur $O(dNM_0)$ computational cost. Additionally, LKRF solves an optimization with $O(M_0 \log \epsilon^{-1})$ time to reach the $\epsilon$-optimal solution, and EERF sorts an array of size $M_0$ with average $O(M_0 \log M_0)$ time. On the other hand, when approximating Gaussian kernel by a second order Taylor expansion, our method forms $O(d^2)$ features and incurs $O(Nd^2)$ computations, which is less than the other two in case $d \ll M_0$. On all data sets except "Year prediction", observe that our method spends drastically smaller pre-processing time to achieve a competitive result after evaluating smaller number of candidate features (i.e., smaller $M_0$). To compare the training cost, if the time cost of the related task (regression or classification) with $M$ features is $\mathcal{C}(M)$, LKRF and EERF simply spend that budget. However, running $K$ iterations of our method (with $M$ a multiple integer of $K$), assuming that repetitive features are not selected, the training cost of MFGA would be $\sum_{k=1}^{K} \mathcal{C}(kM/K)$, which is more than LKRF and EERF. Furthermore, notice that the choice of explicit or random feature maps would too affect the training time. For example, in regression, this directly governs the condition number of the $M \times M$ matrix that is to be inverted. As a result, there exist hidden constants in $\mathcal{C}$ that are different across algorithms. Overall, looking at the sum of training and pre-processing time from Table 2a-2b, we observe that our algorithm can achieve competitive results by spending less time compared to data-dependent methods. For example, on "Online news",

we reduce the error of EERF from $1.63\%$ to $0.57\%$ ($\approx 65\%$ decrease) in $\frac{1.22+0.92}{10.6+0.15}$ time ratio ($\approx 80\%$ decrease).

In general, the comparison of our method to LKRF and EERF is equivalent to the comparison of (data-dependent) explicit-vs-randomized feature maps. In comparison of vanilla (data-independent) explicit-vs-randomized feature maps, as discussed in the experiments of [6] for Gaussian kernel, the performance of none clearly dominates the other. Essentially, Gaussian kernel can be (roughly) seen as (a countable) sum of polynomial kernels as well as (an uncountable) sum of cosine feature maps. Our theoretical bound, which holds for countable sums, suggests that for "good" explicit feature maps, the coefficients may vanish fast (small $\mathcal{E}_{\text{spec}}$), i.e., there exists a *sparse* representation, but of course, such feature map is *unknown* before the learning process.

**Comparison with kernel methods:** As we observe in Table 2a-2b, our method outperforms GK and GLK on "Year prediction" and "Adult". For "Year prediction", our $t_{pp} + t_{\text{train}}$ divided by the training time of GK is $(5.33 + 4.25)/139.5 \approx 0.068$. The same number for "Adult" is $\approx 0.036$, exhibiting a dramatic decrease in the runtime. Noticing that (except for "Epileptic seizure recognition") we used a subsample of training data for kernel methods (due to computational cost), the actual runtime decrease is even more remarkable (2 to 3 orders of magnitude). For "Online news popularity" and "Epileptic seizure recognition", our method is outperformed in terms of accuracy but still saves significant computational cost while being competitive to kernel methods.

Table 2: Comparison of the error and time cost of our algorithm versus other baselines. $M_0$ is the number of candidate features and $M$ is the number of features used for training and testing. $t_{pp}$ and $t_{\text{train}}$, respectively, represent pre-processing and training time (seconds). For kernel methods, we use a subsample $N_0$ of the training set. For all methods, the test error (%) is reported with standard errors in parentheses for randomized approaches.

(a) Results on regression: Year prediction (left) and Online news (right)

| Method | $M$ | $M_0$ | $N_0/N$ | $t_{pp}$ | $t_{\text{train}}$ | error (%) | Method | $M$ | $M_0$ | $N_0/N$ | $t_{pp}$ | $t_{\text{train}}$ | error (%) |
|---|---|---|---|---|---|---|---|---|---|---|---|---|---|
| RKS | 400 | – | – | – | 0.63 | 8.27 (4e-2) | RKS | 200 | – | – | – | 0.13 | 3.08 (5e-2) |
| LKRF | 400 | 4000 | – | 3.5 | 0.62 | 8.51 (8e-2) | LKRF | 200 | 20000 | – | 9.8 | 0.14 | 2.07 (5e-2) |
| EERF | 400 | 4000 | – | 3.3 | 0.64 | 8.76 (6e-2) | EERF | 200 | 20000 | – | 10.6 | 0.15 | 1.63 (4e-2) |
| This work | 400 | 4186 | – | 5.33 | 4.25 | **4.78** | This work | 200 | 1770 | – | 1.22 | 0.92 | 0.57 |
| GK | – | – | 0.5 | – | 139.5 | 5.7 | GK | – | – | 0.3 | – | 240.9 | 0.23 |
| GLK | – | – | 0.5 | – | 150.6 | 5.08 | GLK | – | – | 0.3 | – | 257.6 | **0.14** |

(b) Results on classification: Adult (left) and Epileptic seizure recognition (right)

| Method | $M$ | $M_0$ | $N_0/N$ | $t_{pp}$ | $t_{\text{train}}$ | error (%) | Method | $M$ | $M_0$ | $N_0/N$ | $t_{pp}$ | $t_{\text{train}}$ | error (%) |
|---|---|---|---|---|---|---|---|---|---|---|---|---|---|
| RKS | 100 | – | – | – | 0.87 | 17.7 (6e-2) | RKS | 20 | – | – | – | 0.06 | 6.21 (9e-2) |
| LKRF | 100 | 2000 | – | 1.4 | 0.91 | 16.46 (3e-2) | LKRF | 20 | 2000 | – | 4.2 | 0.06 | 5.24 (4e-2) |
| EERF | 100 | 2000 | – | 2 | 1.38 | 16.15 (2e-2) | EERF | 20 | 2000 | – | 6.8 | 0.07 | 4.46 (4e-2) |
| This work | 100 | 245 | – | 0.19 | 0.69 | **15.10** | This work | 20 | 357 | – | 0.08 | 0.32 | 4.73 |
| GK | – | – | 0.25 | – | 24.07 | 15.70 | GK | – | – | 1 | – | 12.95 | **2.82** |
| GLK | – | – | 0.25 | – | 77.09 | 15.22 | GLK | – | – | 1 | – | 73.02 | 3.41 |

# Acknowledgements

We gratefully acknowledge the support of DARPA Grant W911NF1810134.

## Footnotes

[1]In this paper, our focus is on "explicit features", and whenever it is clear from the context, we simply use "features" instead.

[2]A symmetric function $K : \mathcal{X} \times \mathcal{X} \to \mathbb{R}$ is positive semi-definite if $\sum_{i,j=1}^{N} \alpha_i \alpha_j K(\mathbf{x}_i, \mathbf{x}_j) \geq 0$ for $\boldsymbol{\alpha} \in \mathbb{R}^N$.

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
