[Reviews · NeurIPS 2018]

Reviewer 1



### Post-Rebuttal ### I have read the rebuttal. Thank you very much for the clarifications. ################## # Summary The paper builds on [1] and proposes an algorithm for feature selection from an a priori specified finite dictionary/set of features. In particular, [1, Algorithm 3] propose an approach for minimization of the expected loss of a linear predictor that aims at finding a `good' sparse solution. The main idea of the algorithm from [1] is to iteratively add features by picking a previously unselected feature that amounts to the largest reduction in the expected risk. Then, a linear model is trained using the extended feature representation and afterwards the whole process is repeated. The authors use pretty much the same idea and take a large dictionary of features to represent the data. Following this, they run Algorithm 3 from [1] to pick `informative' features and generate a sparse feature representation. In particular, the step 2 from Algorithm 1 proposed by the authors can be shown to be `almost equivalent' to the first step of Algorithm 3 in [1]. The approach is motivated by the problems with scalability of kernel methods and related to learning with multiple kernels. In particular, a positive definite Mercer kernel can be represented as an inner product of possibly countably infinitely many features/eigenfunctions. For multiple kernels, it is possible to concatenate the feature representations corresponding to different kernels. For a Mercer kernel, the authors aim to find subsets of the features that produce a good approximation of the inner product defined by it. There are two theoretical results: i) A bound on the Rademacher complexity of the hypothesis space defined by a dictionary of pre-selected features. This result builds on [3] and [4] and it is a rather straightforward derivation, following the previous results. ii) A bound on the expected risk of the estimator generated by the proposed algorithm. There are two terms that are bounded when deriving this result. The first term bounds the difference between the risk of an estimator generated by Algorithm 1 and the best estimator over the hypothesis space defined by a dictionary of features. The most important result for this term follows from [1]. The second term bounds the difference between the expected risk of the best estimator over the hypothesis space defined by the dictionary and that over the combination of multiple kernels with possibly infinitely many features. To the best of my knowledge, this piece is novel and interesting. Here it is important to note that similar bounds (finite vs. infinite dictionary) have been considered in [5] for random Fourier features. The bound on the expected risk consists of three terms: i) a sample complexity term accounting for the concentration with respect to the number of instances (the standard \sqrt{n} rate), ii) an approximation term that accounts for the effect of adding more features and extending the feature representation (approximation properties improve exponentially, quite interesting), and iii) a spectral property term that accounts for the coefficient mass present over the (possibly infinitely many) remaining features defined on the whole feature space. The empirical evaluation is performed using a dictionary of random Fourier features and the approach is compared to 5 baselines. The first baseline is the random Fourier feature approximation of the Gaussian kernel. The second and the third baselines sample a large pool of features and then perform feature selection using a scoring function and kernel alignment, respectively. The fourth approach is learning with the standard Gaussian kernel and the fifth one is multiple kernel learning with a combination of Gaussian and linear kernels. Overall the results show an improvement over the competing baselines and good test prediction times. # Comments - The paper is well-written and easy to follow. The derivations are complete and understandable. The provided proofs are also legible and cite all the auxiliary results from previous work. - I recognize that there is a subtle difference between Algorithm 1 proposed by the authors and Algorithm 3 from [1]. In particular, [1] consider feature selection with respect to the expected risk and state that the result holds for the empirical risk, as well. The authors, on the other hand, consider empirical risk when picking a new feature and this makes their result from Theorem 2 novel and interesting. - Related to the previous point, Algorithm 1 should include a regularization term with a hyperparameter controlling the model complexity? - A limitation of the paper is the fact that the total number of features in the dictionary needs to be finite. A number of approaches have been developed for boosting features that work with infinite dictionaries [6-10]. I also think that approaches considered in [6, 8, 10] are quite related to the goal of this paper. Moreover, it would be very interesting to compare the effectiveness of feature selection to an approach like [9]. For a dictionary of features selected by sampling from a Gaussian distribution with thick tails the feature selection could be competitive with [9], where a mixture of Gaussian distributions models the spectral density of a stationary kernel and outputs a feature representation. - The spectral term from the bound in Theorem 2 deserves some more space. I do see that the spectral decay of the kernel can affect the eigenvalue decay but in this particular case we have coefficients of the model. It would be instructive to assume some decay (e.g., polynomial or exponential) and produce an estimate or rate for this term (this should be related to the remark on the Parseval's theorem). - Related to the previous point, the paper makes a strong assumption that a large dictionary of features specified a priori covers the `important' coefficients/features. In general, this is quite difficult to satisfy and the bound indicates that the concentration might not happen at all if the dictionary is not well-equipped to model the conditional dependence of interest. - A minor issue with the empirical evaluation is the fact that the bandwidth parameter was fixed and not cross-validated. The gap between the proposed approach and other baselines might be `over emphasized' as a result of this. - In Table 2, why is there such a large difference in test time predictions between LKRF, EERF, and the proposed approach? Are they not predicting with the same number of features? - Line 97, citation is needed for Parseval's theorem. # References +[1] S. Shalev-Shwartz, N. Srebro, T. Zhang -- Trading accuracy for sparsity in optimization problems with sparsity constraints (SIAM Journal of Optimization, 2010) +[2] M. Mohri, A. Rostamizadeh, A. Talwalkar -- Foundations of machine learning (MIT Press, 2012) +[3] P. Bartlett, S. Mendelson -- Rademacher and Gaussian complexities: risk bounds and structural results (JMLR, 2002) +[4] C. Cortes, M. Mohri, A. Rostamizadeh -- Generalization bounds for learning kernels (ICML, 2010) --- -[5] A. Rahimi, B. Recht -- Uniform approximation of functions with random bases (IEEE, 2008) -[6] F. Huang, J. T. Ash, J. Langford, R. E. Shapire -- Learning deep ResNet blocks sequentially using boosting theory (ICML, 2018) -[7] C. Cortes, M. Mohri, U. Syed -- Deep boosting (ICML, 2014) -[8] C. Cortes, X. Gonzalvo, V. Kuznetsov, M. Mohri, S. Yang -- AdaNet: adaptive structural learning of artificial neural networks (ICML, 2017) -[9] Z. Yang, A. Smola, L. Song, A. G. Wilson -- A la carte-learning fast kernels (AISTATS, 2015) -[10] D. Oglic, T. Gaertner -- Greedy feature construction (NIPS, 2016)

Reviewer 2



After reading the rebuttal: the authors addressed my concerns about orthogonality (a misunderstanding on my part) and the other minor issues I pointed out. My score of accept remains the same. This work presents a multiple kernel learning approach to selecting explicit nonlinear features to approximately solve kernelized empirical risk minimization problems. This work is part of a line of recent work--- motivated by the promise of random feature maps to reduce the computational complexity of kernel methods--- to find low-dimensional explicit representations of hypothesis functions that are competitive with the implicit representations that rely on the computationally unwieldy kernel matrix. The novelty of the work lies in that unlike most recent approaches, it presents a deterministic, rather than randomized, approach to selecting these explicit features. It does so by framing the problem as one of greedily selecting basis functions from each of several hypothesis RKHS spaces to reduce the current residual error. The authors show empirically that this approach can beat (in terms of test error), prior randomized algorithms for generating explicit features. Strengths: - The framing of the problem as an OMP-like problem is new as far as I am aware, and may open a fruitful area for further work - The paper is clearly written - The theoretical analysis looks reasonable (I did not read the supplementary material) Weaknesses: - This is not actually an OMP algorithm as the authors claim, because their algorithm selects for multiple orthogonal bases, instead of just one. This is worth mentioning since it means the intuition that we have from the analysis of OMP, that it will converge to a minimum, does not hold here - The authors analyze their algorithm assuming that the basis functions are orthonormal, but it is not clear that the experiments conformed to this assumption: are the basis functions obtained from the Taylor Series expansion of the gaussian kernel orthonormal? The authors should make it clear either way. - The theory is provided for hard constraints on the mixture coefficients, while the experimental results are given for a lagrangian relaxation. It is possible to obtain theory that applies directly to the lagrangian relaxation - I would like to have seen a comparison with a fast randomized feature map such as FastFood: the authors compare to the basic random Fourier Feature map, which is much slower because it uses more iid random variables. I believe the practice is to use FastFood. - The graphs are illegible when the paper is printed in black and white. It would be preferable to use markers and different line styles to help avoid this issue.

Reviewer 3



Summary: This paper presents a greedy feature selection algorithm (called MFGA) which sequentially selects features from a set of pre-defined kernel features. A generalization guarantee is given for the proposed algorithm. MFGA is shown to give better test performance than several kernel feature selection/reweighing algorithms on real datasets. Strength: Overall the paper is well-written and easy to follow. The proposed algorithm is simple to implement. Weakness: - The proof of Theorem 2 directly uses a previous result (i.e., Theorem 3 in the supplementary) given by Shalev-Shwartz, Srebro and Zhang’2010 (denoted as [SSZ10] in the following). However, there is a l2 norm constraint in the proposed MFGA algorithm, while there is no such constraint in the corresponding algorithm (the Fully Corrective Forward Greedy Selection Algorithm) proposed in [SSZ10]. The author should give justification of directly using the result in [SSZ10] to the proposed MFGA algorithm. Intuitively, the convergence rate of MFGA should also depend on the l2 norm constraint, however, the current result (i.e., Theorem 3 in the supplementary) does not have such dependence. - Important references seems missing. Not sure if the authors are aware of the following related papers: Ian E.H Yen et al., “Sparse random features algorithm as coordinate descent in Hilbert space”, NIPS 2014; Brian Bullins et al., “Not-So-Random Features”, ICLR 2018. In particular, the empirical performance of method proposed in the “Not-So-Random Features” paper is claimed to be much better than the baseline algorithm LKRF chosen in the paper. --------------------------After reading the rebuttal------------------------- I read the authors’ response. The authors have addressed my comments. Specifically, one of my concerns is on the missing dependency of the l2-norm bound in the proof of Thm2. The authors point it out this is because they *assume* that the optimal solution in each iteration of MFGA always lies in the interior of the l2 ball. Besides, the authors mentioned that the recent ICLR paper focuses on classification while this paper focuses on regression. It would be interesting to see how the two methods compare over classification tasks. Overall, I am happy to increase my score.